# Mixed-Memory RNNs for Learning Long-term Dependencies in Irregularly-sampled Time Series

## Abstract

Recurrent neural networks (RNNs) with continuous-time hidden states are a natural fit for modeling irregularly-sampled time series. These models, however, face difficulties when the input data possess long-term dependencies. We prove that similar to standard RNNs, the underlying reason for this issue is the vanishing or exploding of the gradient during training. This phenomenon is expressed by the ordinary differential equation (ODE) representation of the hidden state, regardless of the ODE solver's choice. We provide a solution by equipping arbitrary continuous-time networks with a memory compartment separated from their time-continuous state. This way, we encode a continuous-time dynamical flow within the RNN, allowing it to respond to inputs arriving at arbitrary time-lags while ensuring a constant error propagation through the memory path. We call these models Mixed-Memory-RNNs (mm-RNNs). We experimentally show that Mixed-Memory-RNNs outperform recently proposed RNN-based counterparts on non-uniformly sampled data with long-term dependencies.

## 1 Introduction

Irregularly-sampled time series, routine data streams in medical and business settings, can be modeled effectively by a time-continuous version of recurrent neural networks (RNNs). These class of RNNs whose hidden states are identified by ordinary differential equations, termed an ODE-RNN (Rubanova et al., 2019), provably suffer from the vanishing and exploding gradient problem (see Figure 1, the first two models), when trained by reverse-mode automatic differentiation (Rumelhart et al., 1986; Pontryagin, 2018).

An elegant solution to the vanishing gradient phenomenon (Hochreiter, 1991; Bengio et al., 1994), which results in difficulties in learning long-term dependencies in RNNs, is memory gating. This technique, first published in form of the long short term memory networks (LSTM) (Hochreiter & Schmidhuber, 1997), enforce a constant error propagation through the hidden states, learn to forget, and disentangle the hidden states (memory) from their output states. Despite becoming the standard choice in modeling regularly-sampled temporal dynamics, memory gated RNNs similar to other discretized RNN models, face difficulties when the time-gap between the observations are irregular.

In this paper, we propose a compromise to design a novel recurrent neural network algorithm that simultaneously enjoys the approximation capability of ODE-RNNs in modeling irregularly-sampled time series and capability of learning long-

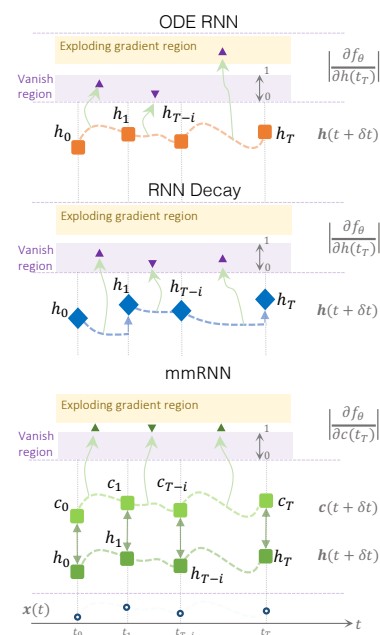

Figure 1: Magnitude of the states' error propagation in time-continuous recurrent neural networks gives rise to the vanishing or exploding of the gradient (first two models). mmRNNs avoid these phenomena in modeling irregularly sampled data.

term dependencies of the memory gated RNN's computational graph. To perform this, we let a memory cell compute its implicit memory mechanism by their typical (input, forget, and output) gates while receiving their feedback inputs from a time-continuous output state representation. This way, we incorporate a continuous-time dynamical flow within a gated recurrent module (e.g., an LSTM), enabling cells to respond to data arriving at arbitrary time-lags, while avoiding the vanishing gradient problem. Since these models interweave continuous flows with discrete memory mechanisms, we call them mixed-memory RNNs a (See Figure 1, the last model).

We compare mmRNNs to standard and advanced continuous-time RNN variants, on a set of synthetic and real-world sparse time-series tasks, and discover consistently better performance.

To put this in context, we first theoretically prove that the class of ODE-RNNs suffers from the exploding and vanishing gradient problem, making them unable to learn long-term dependencies efficiently. We show that learning ODE-RNNs by the adjoint method (Chen et al., 2018) does not help with this problem. As a solution, we propose mmRNNs, a continuous-time recurrent model capable of learning long-term dependencies of irregularly-sampled time-series.

## 2 Background

**ODE-RNNs** Instead of explicitly defining a state update function, ODE-RNNs identify an ordinary differential equations in the following form (Funahashi & Nakamura, 1993):

$$\frac{\partial h}{\partial t} = f_\theta(x_{t+T}, h_t, T) - \tau h, \tag{1}$$

where $x_t$ is the input sequence, $h_t$ is an RNN's hidden state, and $\tau$ is a dampening factor. The time-lag $T$ specifies at what times the inputs $x_t$ have been sampled.

ODE-RNNs were recently rediscovered (Rubanova et al., 2019) and have shown promise in approximating irregularly-sampled data, thanks to the implicit definition of *time* in their resulting dynamical systems. ODE-RNNs can be trained by backpropagation through time (BPTT) (Rumelhart et al., 1986; Werbos, 1988; 1990) through ODE solvers, or by treating the solver as a black-box and apply the adjoint method (Pontryagin, 2018) to gain memory efficiency (Chen et al., 2018). In Section 3, we show this family of recurrent networks faces difficulty to learn long-term dependencies.

**Long Short-term Memory** LSTMs (Hochreiter & Schmidhuber, 1997) express their discretized hidden states as a pair $(c_t, h_t)$ and its update function realizes a mapping as follows: $f_\theta(x_{t+1}, (c_t, h_t), 1) \mapsto (c_{t+1}, h_{t+1})$.

LSTMs demonstrate great performance on learning equidistant streams of data (Greff et al., 2016), however similar to other discrete-state RNNs, they are puzzled with the events arriving in-between observations. In Section 4, we introduce a continuous-time long short-term memory algorithm to tackle this.

## 3 ODE-RNNs suffer from vanishing or exploding gradient.

In this section, we show that ODE-RNNs trained via backpropagation through time (BPTT) are susceptible to vanishing and exploding gradients. We also illustrate that the adjoint method is not immune to these gradient issues. We first formally define the gradient problems of the RNNs, and progressively construct Theorem 1.

**Gradient propagation in recurrent networks.**

Hochreiter (Hochreiter, 1991) discovered that the error-flow in the BPTT algorithm realizes a power series that determines the effectiveness of the learning process (Hochreiter, 1991; Hochreiter & Schmidhuber, 1997; Bengio et al., 1994; Pascanu et al., 2013). In particular, the state-previous state Jacobian of an RNN:

$$\frac{\partial h_{t+T}(x_{t+T}, h_t, T)}{\partial h_t}, \tag{2}$$

governs whether the propagated error exponentially grows (explodes), exponentially vanishes, or stays constant. Formally:

**Definition 1** (Per-unit vanishing or exploding gradient). *Let $h_{t+T} = f(x_{t+T}, h_t, T)$ be a recurrent neural network, then we say unit $i$ of the network $f$ suffers from a vanishing gradient if for some small $\varepsilon > 0$ it hold that $\left| \sum_{j=1}^{N} \frac{\partial h_{t+T}^i}{\partial h_t^j} \right| < 1 - \varepsilon$, where $N$ is the dimension of the hidden state $h_t$ and super-script $v^i$ denotes the $i$-th entry of the vector $v$. We say unit $i$ of the network $f$ suffers from an exploding gradient if it holds that $\left| \sum_{j=1}^{N} \frac{\partial h_{t+T}^i}{\partial h_t^j} \right| > 1$. We say the whole network $f$ suffers from a vanishing or respectively exploding gradient problem if the above condition hold for some of its units.*

The factor $\varepsilon$ in Definition 1 is essential as Gers et al. (Gers et al., 2000) observed that a learnable vanishing factor in the form of a forget-gate significantly benefits the learning capabilities of RNNs, i.e., the network can *learn to forget*. Note that a RNN can simultaneously suffer from a vanishing and an exploding gradient by the definition above.

Now, consider an ODE-RNN given by Eq. 1 is implemented either by an Explicit Euler discretization or by a Runge-Kutta method (Runge, 1895; Dormand & Prince, 1980). We can formulate their state-previous state Jacobian in the following two lemmas:

**Lemma 1.** *Let $\dot{h} = f_\theta(x, h, T) - h\tau$ be an ODE-RNN. Then state-previous state Jacobian of the explicit Euler is given by the following equation: $\frac{\partial h_{t+T}}{\partial h_t} = I + T \frac{\partial f}{\partial h}\Big|_{h=h_t} - \tau T I$.*

**Lemma 2.** *Let $\dot{h} = f_\theta(x, h, T) - h\tau$ be an ODE-RNN. Then state-previous state Jacobian of the Runge-Kutta method is given by $\frac{\partial h_{t+T}}{\partial h_t} = I + T \sum_{j=1}^{M} b_i \frac{\partial f}{\partial h}\Big|_{h=K_i} - \tau T I.$, , where $\sum_{j=1}^{M} b_i = 1$ and some $K_i$.*

The proofs for Lemma 1 and Lemma 2 is provided in the supplements. Consequently, we have:

**Theorem 1. (ODE-RNNs suffer from a vanish or exploding gradient)** *Let $\dot{h} = f_\theta(x, h, T) - h\tau$, and $h_t$ the RNN obtained by simulating the ODE by a solver based on the explicit Euler or Runge-Kutta method. Then the RNN suffers from a vanishing and exploding gradient problem, except for parameter configurations which give the non-trainable constant dynamics $f_\theta(h, x) = 0$, and cases where $f_\theta(h, x)$ is constant, for a particular input sequence $x$ and $\theta$.*

The proof is given in full in the supplementary materials. A brief outline of the proof is as follow: First, we look a the special cases of $\frac{\partial f}{\partial h} - \tau = 0$. While such $f$ would enforce a constant error propagation by making the Jacobians equal to the identity, it also removes all dynamics from the ODE state. In other words, it would operate the ODE as a memory element. Intuitively, any interesting function $f_\theta$ pushes the Jacobians away from the identity matrix, creating a vanishing or exploding gradient depending on $f_\theta$.

Our proof sketch implies a minor but nonetheless interesting statement:

**Corollary 1.** *A ODE-RNN with identity Jacobian matrix expresses the trivial dynamics $\frac{\partial f}{\partial h} - \tau = 0$.*

The proof follows directly by Lemma 1 and 2.

Corollary 1 suggests that well-conditioned gradients of an ODE-RNN can negatively impact the modeling capacity of these models and prevent them from learning the dynamics of the training data. In particular, this result indicates that strong long-term learning results of ODE-based architectures from the literature may be attributed to a better initialization, i.e., with stable gradients being only present at the beginning of the training.

**Theorem 2. (ODE-RNNs suffer from vanishing/exploding gradients regardless of their choice of ODE-solver)** *Let $\dot{h} = f_\theta(x, h, T) - h\tau$, with $f_\theta$ being uniformly Lipschitz continuous. Moreover, let $h(t)$ be the solution of the initial value problem with initial state $h_0$. Then, the gradients $\frac{\partial h(T)}{\partial h_0}$, i.e, the Jacobian of the ODE state at time $T$ w.r.t. the initial state $h_0$, can vanish and explode, except for parameter configurations which give rise to the non-trainable constant dynamics $f_\theta(h, x) = 0$, and cases where $f_\theta(h, x)$ constant, for a particular input sequence $x$ and parameters $\theta$.*

The proof is given in full in the supplementary materials. A brief outline of the proof is as follow: We start by approximating the initial-value problem by an explicit Euler method with a uniform step-size. We then

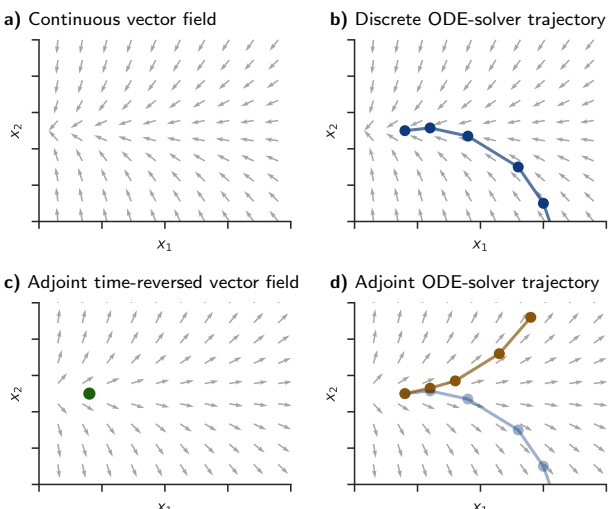

Figure 2: The adjoint method makes numerical error when computing the gradients. **a)** Continuous vector field implied by an ODE. **b)** Numerical ODE-solvers realize a discrete trajectory on the vector field. **c)** The adjoint ODE creates a time-reversed vector field. **d)** Discrete trajectory of the adjoint ODE-solver diverges from the trajectory of the forward simulation due to discretization and numerical imprecision.

let the step-size approach zero which makes the series converge to the true solution of the ODE. Based on bounds on $\frac{\partial f}{\partial h}$, we can obtain bounds of the gradients in the limit, which can vanish or explode depending on $f_\theta$.

Note that the Theorems 1 and 2 hold independently of the used RNN architecture. In particular, we can entail the following statement:

**Corollary 2.** *ODE-combined-with-GRU and ODE-combined-with-LSTM as presented in (Rubanova et al., 2019), also suffer from vanish or exploding gradients.*

The proof follows by applying the Theorems 1 and 2 with a GRU and LSTM.

**Does the adjoint method solve the vanishing gradient problem?** Adjoint sensitivity method (Pontryagin, 2018) allows for performing memory-efficient reverse-mode automatic differentiation for training neural networks with their hidden states defined by ODEs (Chen et al., 2018). The method, however, possesses *lossy* reverse-mode integration steps, as it forgets the computed steps during the forward-pass (Zhuang et al., 2020). Consequently, at each reverse-mode step, the backward gradient pass diverges from the true forward pass (Zhuang et al., 2020; Gholami et al., 2019).

This is because the auxiliary differential equation in the adjoint sensitivity method, $a(t)$, still contains state-dependent components at each reverse-step, which depends on the historical values of the hidden states' gradient. In the extreme case, reverse-steps completely diverge from the hidden states of the forward-time solution, resulting in incorrect gradients. Therefore, both vanilla BPTT and the adjoint method face difficulties for learning long-term dependencies. n the next section, we propose a solution.

## 4   Mixed-Memory Recurrent Architectures

Instead of having a single state vector $h_t$ that is processed by the discrete RNN and the time-continuous ODE, our Mixed-Memory architecture represents its hidden state by a pair $(c_t, h_t)$. An update of the form

$$c_{t+1} = c_t \odot \sigma(g_\theta(h_t) + b_f) + z_\theta(h_t), \tag{3}$$

governs the memory cell component $c_t$ of the mmRNN, where $g_\theta$ and $z_\theta$ are learnable gating functions and $b_f$ a bias term. Although , the update in Eq. (3) appears similar to that of a vanilla LSTM and GRU, there

---

**Algorithm 1** The mixed-memory RNN

**Input:** Datapoints and their timestamps $\{(x_t, t_i)\}_{i=1...N}$
**Parameters:** Weights $\theta$, output weight and bias $W_{output}, b_{output}$
$h_0 = \mathbf{0}$ {ODE state}
$c_0 = \mathbf{0}$ {Memory cell}
**for** $i = 1 \ldots N$ **do**
    $c_i = c_{i-1} \odot \sigma(g_\theta(h_{i-1}, x_i) + b_f) + z_\theta(h_{i-1}, x_i)$ {Memory cell update}
    $h_i = \text{ODESolve}(f_\theta, c_i, t_i - t_{i-1})$ {Time-continuous state update}
    $o_i = h_i W_{output} + b_{output}$
**end for**
**Return** $\{o_i\}_{i=1...N}$

---

Table 1: Change to the hidden states of an RNN between two observations $t$ and $t + T$

| Model | State between observation |
|---|---|
| Standard RNN | $h_t$ |
| GRU-D | $h_t e^{-T\tau}$ |
| ODE-RNN | $\text{ODE-Solve}(f_\theta, h_t, T)$ |
| mmRNN | $\big(c_t, \text{ODE-Solve}(f_\theta, c_t, T)\big)$ |

is one key difference: The gates of a LSTM/GRU is controlled by its previous hidden state, whereas the gating of a mmRNN is defined by a second process $h_t$.

The second part of the hidden state $h_t$ is controlled by a continuous-time process, e.g. a Neural ODE, initialized by the memory cell vector $c_t$. In particular, the update function for $h_t$ is of the form

$$h_{t+1} = \text{ODE-Solve}(f_\theta, c_t, T). \tag{4}$$

The fundamental distinction of mmRNN to other continuous-time RNNs is the strict separation of memory and time-continuous hidden states. In particular, the memory update in Eq. (4) ensures a constant error propagation through $c_t$, while arbitrary Neural ODEs process the state $h_t$ in a time-countinuous fashion.

On the contrary, recurrent network variants such as continuous-time gated recurrent units (CT-GRU) (Mozer et al., 2017), and GRU-D (Che et al., 2018) incorporate a time-dependent decay apparatus on the state, while preserving the rest of the RNN architecture. This decaying memory originates the vanishing factor during backward error-propagation, which results in difficulties in learning long-term dependencies.

Our mmRNNs are immune to this shortcoming. More precisely, Table 1 lists how the transition of the hidden states between two observations of the mmRNN differs from other architectures. Similar to the vanilla LSTM and GRU for regularly sampled time-series, we can ensure a near-constant error propagation at the beginning of the training process with a proper weight initialization.

**mmRNNs allow controlling the vanishing gradient** Initializing the weights of $g_\theta$ and $z_\theta$ close to 0 avoids the units $c_t$ of the state pair $(c_t, h_t)$ from suffering of a vanishing or exploding gradient at the beginning of the training process.

This is because if $g_\theta$ and $z_\theta$ are close to 0 we can neglect them and get $\left| \sum_{j=1}^{N} \frac{\partial c_{t+T}^i(x_{t+T}, (c_t, h_t), T)}{\partial c_t^j} \right| = \sigma(b_f) \approx 0.9943$, which is less than 1 (no exploding) but much greater than 0 (no vanishing). Note that exact value of the Jacobian at the beginning of the training can be controlled by the bias term $b_f$. If the underlying data express very long-term dependencies, we can increase $b_f$ and bring the error flow factor even closer to 1.

The mmRNN can be viewed as a memory cell with gates controlled by a time-continuous process realized by ordinary differential equations. Next, we evaluate the performance of mmRNNs in multiple time-series prediction tasks.

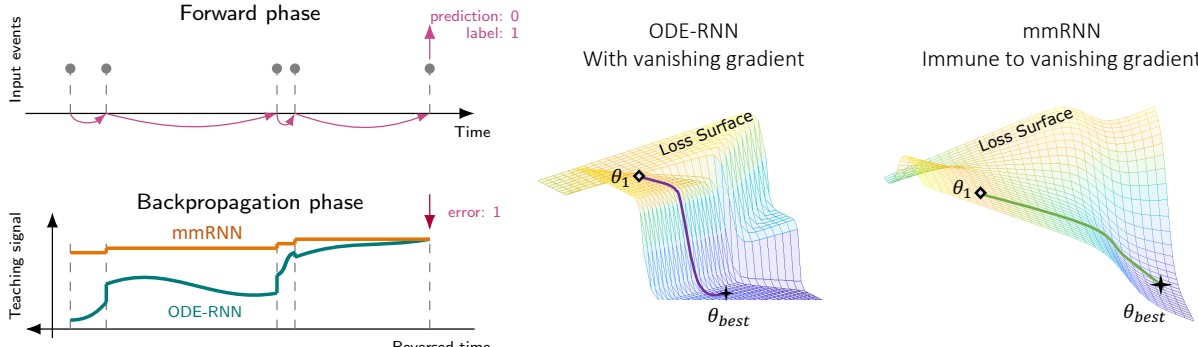

Figure 3: Left: Illustration of how vanishing gradients make RNN training difficult when the data express long-term dependencies. The prediction error can be thought of as a teaching signal indicating how the dynamics should be changed to minimize the loss. The vanishing gradient of the ODE-RNN makes the teaching signal weaker when propagating it back in time. Conversely, the teaching signal stays near-constant in mmRNNs. Right: The resulting loss surfaces of the ODE-RNN is much steeper than mmRNN, making the training difficult.

**Empirical measurement of gradient norms.** To emphasize the impact of our theoretical results, we performed an experiment comparing the hidden state gradient norms of an ODE-RNN instance compared to another ODE-RNN equipped with mmRNN for different sequence lengths (mean/std over 3 initialization seeds). The results show (as proven in Theorem 1 and 2) that the gradients of a standard ODE-RNN tend to exponentially increase with the length of the input sequence. This gradient issue is significantly improved when mmRNNs are used (See Table 2).

Table 2: Gradient Norms ODE-RNN vs mmRNN. (n =3)

| Sequence length | ODE-RNN | mmRNN |
|:---:|:---:|:---:|
| 5 | $10.50 \pm 5.47$ | $0.97 \pm 0.01$ |
| 10 | $38.75 \pm 33.20$ | $0.96 \pm 0.03$ |
| 25 | $235.76 \pm 383.04$ | $1.01 \pm 0.10$ |
| 50 | $1214.94 \pm 3750.62$ | $1.25 \pm 0.25$ |

## 5 Experimental evaluation

We constructed quantitative settings with synthetic and real-world benchmarks. We assessed the generalization performance of time-continuous RNN architectures on datasets that are deliberately created to express long-term dependencies and are of irregularly-sampled nature. All code and data are submitted as a supplement to our manuscript.

**Baselines.** We compare mmRNN to a large variety of continuous-time RNNs introduced to model irregularly-sampled data. This set includes RNNs with continuous-state dynamics such as ODE-RNN (Rubanova et al., 2019) and CT-RNNs (Funahashi & Nakamura, 1993), state-decay mechanisms such as CT-GRU (Mozer et al., 2017), RNN Decay (Rubanova et al., 2019), CT-LSTM (Mei & Eisner, 2017), and GRU-D (Che et al., 2018), in addition to oscillatory models such as Phased-LSTM (Neil et al., 2016), CoRNNs (Rusch & Mishra, 2021), iRNNs (Kag et al., 2019), and Lipschitz RNNs (Erichson et al., 2021). Furthermore, we tested mmRNNs against intuitive time-gap modeling approaches we built here, termed an augmented LSTM topology as well as reciprocal RNNs (Babaei et al., 2010). Experimental settings are given Appendix.

Table 3: **Bit-stream sequence classification. Note:** Test accuracy (mean $\pm$ std, $N = 5$). While all RNNs can represent the correct function, training is difficult due to long-term dependencies.

| Model | Dense encoding | Event-based encoding |
|---|---|---|
| ODE-RNN | 50.47% $\pm$ 0.06 | 51.21% $\pm$ 0.37 |
| CT-RNN | 50.42% $\pm$ 0.12 | 50.79% $\pm$ 0.34 |
| Augmented LSTM | **100.00% $\pm$ 0.00** | 89.71% $\pm$ 3.48 |
| CT-GRU | **100.00% $\pm$ 0.00** | 61.36% $\pm$ 4.87 |
| RNN Decay | 60.28% $\pm$ 19.87 | 75.53% $\pm$ 5.28 |
| Reciprocal RNN | **100.00% $\pm$ 0.00** | 90.17% $\pm$ 0.69 |
| GRU-D | **100.00% $\pm$ 0.00** | 97.90% $\pm$ 1.71 |
| PhasedLSTM | 50.99% $\pm$ 0.76 | 80.29% $\pm$ 0.99 |
| GRU-ODE | 50.41% $\pm$ 0.40 | 52.52% $\pm$ 0.35 |
| CT-LSTM | 97.73% $\pm$ 0.08 | 95.09% $\pm$ 0.30 |
| iRNN | 49.99% $\pm$ 1.20 | 50.54% $\pm$ 0.94 |
| coRNN | **100.00% $\pm$ 0.00** | 52.89% $\pm$ 1.25 |
| Lipschitz RNN | **100.00% $\pm$ 0.00** | 52.84% $\pm$ 3.25 |
| mmRNN (ours) | **100.00% $\pm$ 0.00** | **98.89% $\pm$ 0.26** |

## 5.1 Synthetic benchmark - Bit-stream sequence classification

We formulated a modified time-series variant of the XOR problem (Marvin & Seymour, 1969). In particular, the model observes a block of binary data in the form of a bit-after-bit time-series. The objective is to learn an XOR function of the incoming bit-stream. This setup is equivalent to the binary-classification of the input sequence, where the labels are obtained by applying an XOR function to the inputs.

While any non-linear recurrent neural network architecture can learn the correct function, training the network to do so is non-trivial. For the model to make an accurate prediction, all bits in an upcoming chunk are required to be taken into account. However, the error signal is only provided after the last bit is observed. Consequently, during learning, the prediction error needs to be propagated to the first input time-step to precisely capture the dependencies.

We designed two modes, a dense encoding mode in which the input sequence is represented as a regular, periodically sampled time-series, and an event-based mode which compresses the data into irregularly sampled bit-streams, e.g., $1, 1, 1, 1$ is encoded as $(1, t = 4)$. (See Table 3).

We observed that a considerable number of RNNs face difficulties in modeling these tasks, even in dense-encoding mode. In particular, ODE-RNNs, CT-RNNs, RNN-Decay, Phased-LSTM, and GRU-ODE could not solve the XOR problem in the first mode. Phased-LSTM and RNN-Decay improved their performance in the second modality, whereas ODE-RNNs, CT-RNNs, and GRU-ODE still could not solve the task.

The core reason for their low performance is the exploitation of the vanishing gradient problem during training. The rest of the RNN variants (except CT-GRU) were successful in solving the task in both modes, with mmRNN outperforming others in an event-based encoding scenario.

## 5.2 Person activity recognition with irregularly sampled time-series

We consider the person activity recognition dataset from the UCI repository (Dua & Graff, 2017). This task's objective is to classify the current activity of a person, from four inertial measurement sensors worn on the person's arms and feet. Even though the four sensors are measured at a fixed period of 211ms, the random phase-shifts between them creates an irregularly sampled time-series. Rubanova et al. (Rubanova et al., 2019) showed that ODE-based RNN architectures perform remarkably well on this dataset. Here, we benchmarked the performance of the mmRNN model against other variants.

Table 4: **Per time-step classification**. Person activity recognition. Test accuracy (mean $\pm$ std, $N = 5$)

| Model | Accuracy |
|---|---|
| ODE-RNN | 80.43% $\pm$ 1.55 |
| CT-RNN | 83.65% $\pm$ 1.55 |
| Augmented LSTM | **84.11% $\pm$ 0.68** |
| CT-GRU | 79.48% $\pm$ 2.12 |
| RNN Decay | 62.89% $\pm$ 3.87 |
| Reciprocal RNN | **83.85% $\pm$ 0.45** |
| GRU-D | 83.57% $\pm$ 0.40 |
| PhasedLSTM | 83.33% $\pm$ 0.69 |
| GRU-ODE | 82.56% $\pm$ 2.63 |
| CT-LSTM | **84.13% $\pm$ 0.11** |
| iRNN | 74.56% $\pm$ 1.29 |
| coRNN | 78.89% $\pm$ 0.62 |
| Lipschitz RNN | 81.35% $\pm$ 0.60 |
| mmRNN (ours) | **84.15% $\pm$ 0.33** |

Table 5: **Event sequence classification**. Irregular sequential MNIST. Test accuracy (mean $\pm$ std, $N = 5$)

| Model | Accuracy |
|---|---|
| ODE-RNN | 72.41% $\pm$ 1.69 |
| CT-RNN | 72.05% $\pm$ 0.71 |
| Augmented LSTM | 82.10% $\pm$ 4.36 |
| CT-GRU | 87.51% $\pm$ 1.57 |
| RNN Decay | 88.93% $\pm$ 4.06 |
| Reciprocal RNN | 94.43% $\pm$ 0.23 |
| GRU-D | 95.44% $\pm$ 0.34 |
| PhasedLSTM | 86.79% $\pm$ 1.57 |
| GRU-ODE | 80.95% $\pm$ 1.52 |
| CT-LSTM | 94.84% $\pm$ 0.17 |
| iRNN | 95.51% $\pm$ 1.95 |
| coRNN | 94.44% $\pm$ 0.24 |
| Lipschitz RNN | 95.92% $\pm$ 0.16 |
| mmRNN (ours) | **97.83% $\pm$ 0.37** |

This setting realizes a per-time-step classification problem. That is a new error signal is presented to the network at every time-step which makes the vanishing gradient less of an issue here. The results in Table 4 shows that the mmRNN outperforms other RNN models on this dataset. While the significance of an evaluation on a single dataset is limited, it demonstrates that the supreme generalization ability of mmRNN architecture.

### 5.3 Event-based sequential MNIST

We determined a challenging sequence classification task by designing an event-based version for the sequential-MNIST dataset. For doing this we followed the procedure described below:

1. **Sequentialization + encoding long-term dependencies** — transform the 28-by-28 image into a time-series of length 784
2. **Compression + non-uniform sampling** — encode binary time-series in a event-based format, to get rid of consecutive occurrences of the same binary value, e.g., $1, 1, 1, 1$ is transformed to $(1, t = 4)$. (Read more about this experiment in supplements)

Table 6: **Per time-step regression**. Walker2d kinematic dataset. (mean $\pm$ std, $N = 5$)

| Model | Square-error |
|---|---|
| ODE-RNN | $1.904 \pm 0.061$ |
| CT-RNN | $1.198 \pm 0.004$ |
| Augmented LSTM | $1.065 \pm 0.006$ |
| CT-GRU | $1.172 \pm 0.011$ |
| RNN-Decay | $1.406 \pm 0.005$ |
| Reciprocal RNN | $1.071 \pm 0.009$ |
| GRU-D | $1.090 \pm 0.034$ |
| PhasedLSTM | $1.063 \pm 0.010$ |
| GRU-ODE | $1.051 \pm 0.018$ |
| CT-LSTM | $1.014 \pm 0.014$ |
| iRNN | $1.732 \pm 0.025$ |
| coRNN | $3.241 \pm 0.215$ |
| Lipschitz RNN | $1.781 \pm 0.013$ |
| mmRNN (ours) | $\mathbf{0.883 \pm 0.014}$ |

Using this sequentialization mechansim, we compress the sequences from 784 to padded sequences of 256 irregularly-sampled datapoints. To perform well on this task, RNNs must learn to store some information up to 256 time-steps, while taking the time-lags between them into account. Since an error signal is issued at the end of the sequence, *only an RNN model immune to vanishing gradients can achieve high-degrees of accuracy.*

Table 5 demonstrates that ODE-based RNN architectures, such as the ODE-RNN, CT-RNN, and the GRU-ODE (De Brouwer et al., 2019) struggle to learn a high-fidelity model of this dataset. On the other hand, RNNs built based on a memory mechanism, such as the reciprocal RNN and GRU-D (Che et al., 2018) perform reasonably well, while the performance of mmRNN surpasses others.

### 5.4 Walker2d kinematic simulation

In this experiment, we evaluated how well mmRNNs can model a physical dynamical system. To this end, we collected simulation data of the `Walker2d-v2` OpenAI gym (Brockman et al., 2016) environment using a pre-trained policy The objective of the model was to learn the kinematic simulation of the MuJoCo physics engine (Todorov et al., 2012) in an auto-regressive fashion and a supervised learning modality.

We increased the complexity of this task by using the pre-trained policy at different training stages (between 500 to 1200 Proximal Policy Optimization (PPO) iterations (Schulman et al., 2017)) and overwrote 1% of all actions by random actions. Moreover, we simulated frame-skips by removing 10% of the time-steps. Consequently, the dataset is irregularly-sampled. The results, shown in Table 6, indicate that mmRNNs can capture the kinematic dynamics of the physics engine better than other algorithms with a high margin.

## 6 Related Works

**Time-continuous RNNs** The notion of *CT-RNNs* (Funahashi & Nakamura, 1993) was introduced around three decades ago. It is identical to the ODE-RNN architecture (Rubanova et al., 2019) with an additional dampening factor $\tau$. In our experiments, however, we observed a competitive performance to our mmRNNs achieved by the *GRU-D* architecture (Che et al., 2018).

GRU-D encodes the dependence on the time-lags by a trainable decaying mechanism, similar to *RNN-decay* (Rubanova et al., 2019). While this mechanism enables modeling irregularly sampled time-series, it also introduces a vanishing gradient factor to the backpropagation path.

Similarly, *CT-GRU* (Mozer et al., 2017) adds multiple decay factors in the form of extra dimensions to the RNN state. An attention mechanism inside the CT-GRU then selects which entry along the decay dimension to use for computing the next state update. The CT-GRU aims to avoid vanishing gradients by including a decay rate of 0, i.e., no decay at all. This mechanism nevertheless, fails as illustrated in Table 3.

*Phased-LSTM* (Neil et al., 2016) adds a learnable oscillator to LSTM. The oscillator modulates LSTM to create dependencies on the elapsed-time, but also introduces a vanishing factor in its gradients.

*GRU-ODE* (De Brouwer et al., 2019) modifies the GRU (Chung et al., 2014) topology by incorporating a continuous dynamical system. First, GRU is expressed as a discrete difference equation and then transformed into a continuous ODE. This process makes the error-propagation time-dependent, i.e., the near-constant error propagation property of GRU is abolished.

*Lipschitz RNN* (Erichson et al., 2021) constraints the hidden-to-hidden weight matrix of a continuous-time RNN, such that the underlying dynamic system globally converges to a stable equilibrium. As a result, Lipschitz RNNs cannot suffer from a exploding gradient problem. The constraint of the weight matrix is realized efficiently using a symmetric skew decomposition (Wisdom et al., 2016).

*Log-ODE method* (Morrill et al., 2020) compresses the input time-series by time-continuous path signatures (Friz & Victoir, 2010) before feeding them into ODE-RNNs. As the signatures are much shorter than the original input sequence, the ODE-RNNs can learn long-term dependencies in the original input sequence despite expressing a vanishing gradient.

*CT-LSTM* (Mei & Eisner, 2017) combines the LSTM architecture with continuous-time neural Hawkes processes. At each time-step, the RNN computes two alternative next state options of its hidden state. The actual hidden state is then computed by interpolating between these two hidden states depending on the elapsed time.

*coRNN* (Rusch & Mishra, 2021) uses an implicit-explicit Euler discretization of a second-order ODE modeling a controlled non-linear oscillator. The state-to-next state gradients of the resulting RNN are bounded in both directions, mitigating explosion and vanishing effects.

*iRNN* (Kag et al., 2019) parametrize a RNN by applying incremental updates to the steady-state of an ODE. In the limit with infinitely many updates between the state and the next state, iRNN express a constant error propagation.

*HiPPO* (Gu et al., 2020) presents a framework for continuous-time function memorization. In partiuclar, HiPPO projects the history of time-series to a high order polynomial space and is therefore able to store the history efficiently in the form of coefficients. This mechanisms allows HiPPO-based models Gu et al. (2021b;a) to memorize inputs over extremely long time horizons.

**Learning Irregularly-Sampled Data**     Statistical (Pearson et al., 2003; Li & Marlin, 2016; Belletti et al., 2016; Roy & Yan, 2020) and functional analysis (Foster, 1996; Amigó et al., 2012; Kowal et al., 2019) tools have long been studying non-uniformly-spaced data. A natural fit for this problem is the use of time-continuous RNNs (Rubanova et al., 2019). We showed that although ODE-RNNs are performant models in these domains, their performance tremendously drops when the incoming samples have long-range dependencies. We solved this shortcoming by introducing mmRNNs.

**Learning Long-term Dependencies**     The notorious question of vanishing/exploding gradient (Hochreiter, 1991; Bengio et al., 1994) was identified as the core reason for RNNs' lack of generalizability when trained by gradient descent (Allen-Zhu & Li, 2019; Sherstinsky, 2020). Recent studies used state-regularization (Wang & Niepert, 2019) and long memory stochastic processes (Greaves-Tunnell & Harchaoui, 2019) to analyze long-range dependencies. Apart from the original LSTM model (Hochreiter & Schmidhuber, 1997) and its variants (Greff et al., 2016) that solve the problem in the context of RNNs, very few alternative researches exist (Chen et al., 2019).

As the class of CT RNNs become steadily popularized (Hasani et al., 2020a; Lechner et al., 2020a), it is important to characterize them better (Lechner et al., 2019; Dupont et al., 2019; Durkan et al., 2019) and understand their applicability and limitations (Jia & Benson, 2019; Lechner et al., 2020b; Hanshu et al.,

2020; Holl et al., 2020; Quaglino et al., 2020; Kidger et al., 2020; Hasani et al., 2020b). We proposed a method to enable ODE-based RNNs to learn long-term dependencies.

## 7 Discussions, Scope and Limitations

We proposed a solution to learn long-term dependencies in irregularly-sampled input data streams. To perform this, we designed a novel long short term memory network, that possesses a continuous-time output state, and consequently modifies its internal dynamical flow to a continuous-time model. mmRNNs resolve the vanishing and exploding of the gradient problem of the class of ODE-RNNs while demonstrating an attractive performance in learning long-term dependencies on data arriving at non-uniform intervals.

**What if we feed in samples' time-lag as an additional input feature to network?** The *Augmented LSTM* architecture we benchmarked against realizes this concept, which is a simplistic approach to making LSTMs compatible with irregularly sampled data. The RNN could then learn to make sense of the time input, for instance, by making its change proportional to the elapsed-time. Nonetheless, the time characteristic of an augmented RNN depends purely on its learning process.

Consequently, we can only hope that the augmented RNN generalize to unseen time-lags. Our experiments showed that an augmented LSTM performs reasonably well while being outperformed by models that explicitly declare their state by a continuous-time modality, such as mmRNNs.

**Difference between reciprocal RNNs and mmRNN?** A naive approach of tackling the issues of learning long-term dependencies while also being able to process irregularly sampled time-series is a reciprocal RNN architecture (Babaei et al., 2010). A reciprocal architecture consists of two different types of RNNs reciprocally linked together in an auto-regressive fashion (Babaei et al., 2010). In our context, the first RNN could be designed to handle irregularly-sample time series while the second one is capable of learning long-term dependencies.

For example, an LSTM bidirectionally coupled with an ODE-RNN could, in principle, overcome both challenges. However, the use of heterogeneous RNN architectures might limit the learning process. In particular, due to different training convergence rates, the LSTM could already be overfitting long before the ODE-RNN has learned useful dynamics.

Contrarily, our mmRNN interlinks LSTMs and ODE-RNNs not in an autoregressive fashion, but at an architectural level, avoiding the problem of learning at different speeds. Our experiments showed that mmRNNs consistently outperform a reciprocal LSTM-ODE-RNN architecture.

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

## A  Proofs

**Derivation of the Euler's method Jacobian** Let $\dot{h} = f_\theta(x, h, T) - h\tau$ be an ODE-RNN. Then the explicit Euler's method with step-size $T$ is defined as the discretization

$$h_{t+T} = h_t + T(f_\theta(x, h, T) - h\tau)\Big|_{h=h_t}. \tag{5}$$

Therefore, state-previous state Jacobian is given by

$$\frac{\partial h_{t+T}}{\partial h_t} = I + T\frac{\partial f}{\partial h}\Big|_{h=h_t} - \tau T I. \tag{6}$$

**Derivation of the Runge-Kutta Jacobian** Let $\dot{h} = f_\theta(x, h, T) - h\tau$ be an ODE-RNN. Then the Runge-Kutta method with step-size $T$ is defined as the discretization

$$h_{t+T} = h_t + T\sum_{j=1}^{M} b_i(f_\theta(x, h, T) - h\tau)\Big|_{h=K_i}, \tag{7}$$

where the coefficients $b_i$ and the values $K_i$ are taken according to the Butcher tableau with $\sum_{j=1}^{M} b_i = 1$ and $K_1 = h_t$.

Then state-previous state Jacobian of the Runge-Kutta method is given by the following equation :

$$\frac{\partial h_{t+T}}{\partial h_t} = I + T\sum_{j=1}^{M} b_i\frac{\partial f}{\partial h}\Big|_{h=K_i} - \tau T I., \tag{8}$$

Note that the explicit Euler method is an instance of the Runge-Kutta method with $M = 1$ and $b_1 = 1$.

**Proof of ODE-RNN suffering from vanishing or exploding gradients** Let $\dot{h} = f_\theta(x, h, T) - h\tau$ be an ODE-RNN with latent dimension $N$. Without loss of generality let $h_0$ be the initial state at $t = 0$ and $h_T$ denote the ODE state which should be computed by a numerical ODE-solver. Then ODE-solvers, including fixed-step methods (Runge, 1895) and variable-step methods such as the Dormand-Prince method (Dormand & Prince, 1980), discretize the interval $[0, T]$ by a series $t_0, t_1, \ldots t_n$, where $t_0 = 0$ and $t_n = T$ and each $h_{t_i}$ is computed by a single-step explicit Euler or Runge-Kutta method from $h_{t_{i-1}}$.

Our proof closely aligns with the analysis in Hochreiter and Schmidhuber (Hochreiter & Schmidhuber, 1997). We refer the reader to (Hochreiter, 1991; Bengio et al., 1994; Pascanu et al., 2013) for a rigorous discussion on the vanishing and exploding gradients.

We first prove the theorem for a scalar RNN, i.e., $n = 1$, and then extend the discussion to the general case. The error-flow per RNN step between $t = 0$ and $t = T$ is given by

$$\frac{\partial h_T}{\partial h_0} = \prod_{m=1}^{n} \left(1 + (t_m - t_{m-1})\sum_{j=1}^{M} b_i\frac{\partial f}{\partial h}\Big|_{h=K_{m_i}} - \tau(t_m - t_{m-1})\right), \tag{9}$$

which realizes a power series depending on the value

$$\Big|1 + (t_m - t_{m-1})\sum_{j=1}^{M} b_i\frac{\partial f}{\partial h}\Big|_{h=K_{m_i}} - \tau(t_m - t_{m-1})\Big|. \tag{10}$$

Obviously, the condition that this term is equal to 1 is not enforced during training and violated for any non-trivial $f_\theta$, such as $f_\theta(h, x) = \sigma(W_h h + W_x x + \hat{b})$ with $\sigma$ being a sigmoidal or rectified-linear activation function. The exact magnitude depends on the weights $W_h$, as

$$\frac{\partial f_\theta(h, x)}{\partial h} = W_h \sigma'(W_h h + W_x x + \hat{b}). \tag{11}$$

A non-zero time-constant $\tau$ pushes the gradient toward a vanishing region.

Note that the Equation (10) only becomes equal to 1, if $\sum_{j=1}^{M} b_i \frac{\partial f}{\partial h}\big|_{h=K_{m_i}} = \tau$. This would imply that $\frac{\partial h_{t_m}}{h_{t_m-1}} = 0$, i.e., when the change in ODE-state between two time-points is zero. A variable that does not change over time is a memory element. Thus the only solution of enforcing a constant-error propagation is to include an explicit memory element in the architecture (Hochreiter & Schmidhuber, 1997) which does not change its value between two arbitrary time-points $t_m$ and $t_{m-1}$.

For the general case $n \geq 1$, the error-flow per RNN step between $t = 0$ and $t = T$ is given by

$$\frac{\partial h_T}{\partial h_0} = \prod_{m=1}^{n} \left( I + (t_m - t_{m-1}) \sum_{j=1}^{M} b_i \frac{\partial f}{\partial h}\Big|_{h=K_{m_i}} - \tau(t_m - t_{m-1})I \right). \tag{12}$$

As $h$ is a vector, we need to consider all possible error-propagation paths. The error-flow from unit $u$ to unit $v$ is then given by summing all $N^{n-1}$ possible paths between $u$ to $v$,

$$\frac{\partial h_T^v}{\partial h_0^u} = \sum_{l_1}^{N} \cdots \sum_{l_{n-1}}^{N} \prod_{m=1}^{n} \left( I + (t_m - t_{m-1}) \sum_{j=1}^{M} b_i \frac{\partial f}{\partial h}\Big|_{h=K_{m_i}} - \tau(t_m - t_{m-1})I \right)_{l_m, l_{m-1}}, \tag{13}$$

where $l_0 = u$ and $l_n = v$.

The arguments of the scalar case hold for every individual path in Equation (13). The only difference between the the scalar case and the individual paths in the vectored version is the non-diagonal connections in the general case do not include the constant 1 and $\tau$. The error-propagation magnitude between $u$ and $v$ with $u \neq v$ is given by

$$\left| (t_m - t_{m-1}) \left( \sum_{j=1}^{M} b_i \frac{\partial f}{\partial h}\Big|_{h=K_{m_i}} \right)_{u,v} \right|. \tag{14}$$

Again, for $f_\theta(h, x) = \sigma(W_h h + W_x x + \hat{b})$ we obtain an error-flow that depends on the weights $W_h$ and can be either vanishing or exploding, depending on its magnitude.

**Proof that even gradients of the ODE solution can vanish or explode** Let $\dot{h} = f_\theta(x, h, T) - h\tau$ be an ODE-RNN with latent dimension $N$, with $f_\theta$ being uniformly Lipschitz continuous. Without loss of generality let $h_0$ be the initial state at $t = 0$ and $h_T$ denote the ODE state which should be computed by a numerical ODE-solver. We approximate the interval $[0, T]$ by a uniform discretization grid, i.e. $t_i - t_{i-1} = t_j - t_{j-1} = T/n$ for all $i, j$ $t_0, t_1, \ldots t_n$, where $t_0 = 0$ and $t_n = T$ and each $h_{t_i}$ is computed by a single-step explicit Euler from $h_{t_{i-1}}$.

Even when making the discretization grid $t_0, t_1, \ldots t_n$ finer and finer, the gradient propagation issue is not resolved. Let $h_i$ denote the intermediate values computed by the Picard-iteration, i.e., the explicit Euler. We know that the Picard-iteration $h_T$ converges to the true solution $h(T)$.

First, we assume there exists a $\xi > 0$ such that $\xi \leq \frac{\partial f}{\partial h}\big|_{h=h_m} - \tau$ for all $m$. Note that this situation can naturally occur if we have a $f_\theta(h, x) = \sigma(W_h h + W_x x + \hat{b})$. In the limit $n \to \infty$ we get

$$\lim_{n\to\infty} \frac{\partial h_T}{\partial h_0} = \lim_{n\to\infty} \prod_{m=1}^{n} \left(1 + (t_m - t_{m-1})\frac{\partial f}{\partial h}\Big|_{h=h_m} - \tau(t_m - t_{m-1})\right)$$

$$= \lim_{n\to\infty} \prod_{m=1}^{n} \left(1 + \frac{T}{n}\frac{\partial f}{\partial h}\Big|_{h=h_m} - \tau\frac{T}{n}\right))$$

$$\geq \lim_{n\to\infty} \prod_{m=1}^{n} \left(1 + \frac{T}{n}\xi\right), \text{ with some } 0 < \xi \leq \frac{\partial f}{\partial h}\Big|_{h=h_m} - \tau \text{ for all } m$$

$$= \lim_{n\to\infty} \left(1 + \frac{T}{n}\xi\right)^n$$

$$= e^{T\xi}$$

$$> 1,$$

i.e., we have an exploding gradient.

Conversely, lets assume there exists a $\xi < 0$ such that $\xi \geq \frac{\partial f}{\partial h}\big|_{h=h_m} - \tau$ for all $m$. Note that this situation can also naturally occur, for instance if $\tau > 0$ and regions where $f'$ is small. In the limit $n \to \infty$ we get

$$\lim_{n\to\infty} \frac{\partial h_T}{\partial h_0} = \lim_{n\to\infty} \prod_{m=1}^{n} \left(1 + (t_m - t_{m-1})\frac{\partial f}{\partial h}\Big|_{h=h_m} - \tau(t_m - t_{m-1})\right)$$

$$= \lim_{n\to\infty} \prod_{m=1}^{n} \left(1 + \frac{T}{n}\frac{\partial f}{\partial h}\Big|_{h=h_m} - \tau\frac{T}{n}\right))$$

$$\leq \lim_{n\to\infty} \prod_{m=1}^{n} \left(1 + \frac{T}{n}\xi\right), \text{ with some } 0 > \xi \geq \frac{\partial f}{\partial h}\Big|_{h=h_m} - \tau \text{ for all } m$$

$$= \lim_{n\to\infty} \left(1 + \frac{T}{n}\xi\right)^n$$

$$= e^{T\xi}$$

$$< 1,$$

i.e., we have a vanishing gradient.

Similar to the argument in the proof above, we can extend the scalar case to the general case. However, summing over all possible path might not be trivial, as the number of possible path also growths to infinity.

$$\lim_{n\to\infty} \frac{\partial h_T^v}{\partial h_0^u} = \lim_{n\to\infty} \sum_{l_1}^{N} \cdots \sum_{l_{n-1}}^{N} \prod_{m=1}^{n} \left(I + (t_m - t_{m-1})\frac{\partial f}{\partial h}\Big|_{h=h_m} - \tau(t_m - t_{m-1})I\right)_{l_m, l_{m-1}}. \tag{15}$$

Instead, we assume $u = v = l_1 = \ldots l_n - 1$, i.e., we only look at the error-propagation through the diagonal element $u$.

$$\lim_{n\to\infty} \frac{\partial h_T^v}{\partial h_0^u} = \lim_{n\to\infty} \prod_{m=1}^{n} \left(I + (t_m - t_{m-1})\frac{\partial f}{\partial h}\Big|_{h=h_m} - \tau(t_m - t_{m-1})I\right)_{u,u}$$

$$= \lim_{n\to\infty} \prod_{m=1}^{n} \left(1 + (t_m - t_{m-1})\frac{\partial f^u}{\partial h^u}\Big|_{h^u=h_m^u} - \tau^u(t_m - t_{m-1})\right)$$

$$= \lim_{n\to\infty} \prod_{m=1}^{n} \left(1 + \frac{T}{n}\frac{\partial f^u}{\partial h^u}\Big|_{h^u=h_m^u} - \tau^u\frac{T}{n}\right)),$$

which is equivalent to the scalar case. For an interesting $f$ such as $f_\theta(h, x) = \sigma(W_h h + W_x x + \hat{b})$, the term $\frac{f}{h}$ depends on the value $W_h^{u,u}$. By assuming $W^{w,z}$ for any $(w, z) \neq (u, u)$ is neglectable small, we can infer that the effects of the gradient by any other path in Equation (15) is neglectable small. Thus the global error flow depends on $W_h^{u,u}$, which can make the error-flow either explode or vanish depending on its value.

Note that this argument is similar to arguing that as the multi-dimensional case properly contains the scalar case, the multi-dimensional case can express an exploding or vanishing gradient too.

**A more detailed explanation that mmRNNs do not suffer from a vanishing or exploding gradient at the beginning of the trainng**

Recall that we assume that weights of $g_\theta$ and $z_\theta$ are initialized close to 0 and that we are at the beginning of the training process, i.e., we assume the weights do not differ significantly from their initialized values. Moreover, for the proof we assume that $g_\theta$ and $z_\theta$ are standard multi-layer perceptron modules.

We have

$$
\begin{aligned}
\frac{\partial c_{t+1}}{\partial c_t} &= \frac{\partial c_t \odot \sigma(g_\theta(h_t) + b_f) + z_\theta(h_t)}{\partial c_t} \\
&= \frac{\partial \sigma(g_\theta(h_t) + b_f)}{\partial c_t} \mathrm{diag}(c_t) + \mathrm{diag}(\sigma(g_\theta(h_t) + b_f)) + \frac{\partial z_\theta(h_t)}{\partial c_t}.
\end{aligned}
$$

For the derivatives of the first term we can simply apply the chain-rule and get

$$
\begin{aligned}
\frac{\partial \sigma(g_\theta(h_t) + b_f)^v}{\partial c_t^u} &= \sigma'(g_\theta(h_t) + b_f)^v g_\theta(h_t)^v \frac{\partial g_\theta(h_t)^v}{\partial c_t^u} \\
&\approx 0,
\end{aligned}
$$

because we assumed $g_\theta(h_t) \approx 0$ due to its initialization.

Similar argument holds for the derivative of the last term, where we assumed that $z_\theta$ is initialized closet to 0.

$$
\begin{aligned}
\frac{\partial z_\theta(h_t)^v}{\partial c_t^u} &= \mathrm{f}_{act}'(W_\theta \hat{z}_t + b_\theta) W_\theta \frac{\partial \hat{z}_t^v}{\partial c_t^u} \\
&\approx 0,
\end{aligned}
$$

where $\mathrm{f}_{act}$ is the final activation function of $z_\theta$, $W_\theta$ and $b_\theta$ the weights and bias parametrizing the last layer of $z_\theta$ and $\hat{z}_t$ the last hidden vector of $z_\theta$. Note that we assumed $W_\theta \approx 0$.

Consequently, with a proper weight initialization, the Jacobian simplifies to

$$
\frac{\partial c_{t+1}}{\partial c_t} \approx \mathrm{diag}(\sigma(g_\theta(h_t) + b_f)).
$$

We assumed that $g_\theta$ is initialized close to 0. Hence,

$$
\begin{aligned}
\mathrm{diag}(\sigma(g_\theta(h_t) + b_f))^v &= \approx \sigma(b_f) \\
&= \sigma(3) \\
&\approx 0.994,
\end{aligned}
$$

as we initialized $b_f$ to 3.

Hence, we have

$$
\left| \sum_{j=1}^{N} \frac{\partial c_{t+1}^i}{\partial c_t^j} \right| \approx 0.994,
$$

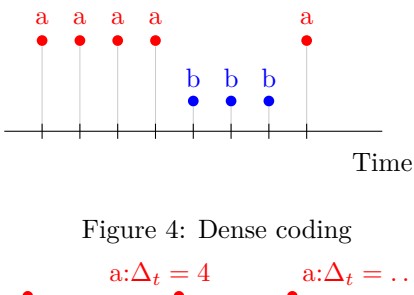

Figure 4: Dense coding

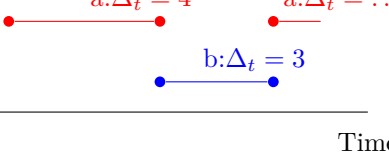

Time

Figure 5: Event-based coding

Figure 6: Dense and event-based coding of the same time-series. An event-based coding is more efficient than a dense coding at encoding sequences where the transmitted symbol changes only sparsely.

, which is less than 1 (no exploding) but much greater than 0 (no vanishing) and ensures a near-constant error propagation at the beginning of the training process.

As already mentioned in the paper, the exact value of the error flow can be controlled by changing the bias term from its default value of 1. If the underlying data distribution contains dependencies with a very long time-lag, we can bring the error flow factor closer to 1 by increasing forget gate bias; Thus enabling the mmRNN to learn even very long-term dependencies in the data.

## B  Experimental evaluation

For models containing differential equations, we used the ODE-solvers as listed in Table 7. Hyperparameter settings used for our evaluation is shown in Table 8.

**Batching** Sequences of our event-based bit-stream classification task and event-based seqMNIST can have different lengths. To allow an arbitrary batching of several sequences, we pad all sequences to equal length and apply a binary mask during training and evaluation.

### B.1  Dataset description

The individual datasets are created as follows:

**Bit-stream XOR dataset** Every data point is a block of 32 random bits. The binary labels are created by applying an XOR function on the bit block, i.e., class A if the number of 1s in the bit-stream are even, class B if the number of 1s in the bit-stream is odd. For training, a cross-entropy loss on these two classes is used. The training set consists of 100,000 samples, which are less than 0.0024% of all possible bit-streams that can occur. The test set consists of 10,000 samples.

For the event-based encoding, we introduce a time-dimension. The time is normalized such that the complete sequence equals 1 unit of time, i.e., 32 bits corresponds to exactly 1 second. An illustration of the two different encodings is shown in Figure 6.

**Person Activity** We consider a variation of the "Human activity" dataset described in (Rubanova et al., 2019) form the UCI machine learning repository (Dua & Graff, 2017). The dataset is comprised of 25 recordings of human participants performing different physical activities. The eleven possible activities are "lying down", "lying", "sitting down", "sitting", "standing up from lying", "standing up from", "sitting", "standing up from sitting on the ground", "walking", "falling", "on all fours", and "sitting on the ground".

The objective of this task is to recognize the activity from inertial sensors worn by the participant, i.e., a per-time-step classification problem. We group the eleven activities listed above into seven different classes, as proposed by (Rubanova et al., 2019).

The input data consists of sensor readings from four inertial measurement units placed on the participant's arms and feet. The sensors are read at a fixed period of 211 ms but have different phase-shifts in the 25 recordings. Therefore, we treat the data as irregularly sampled time-series.

The 25 recordings are split into partially overlapping sequences of length 32, to allow an efficient training of the machine learning models.

Our results are not directly comparable to the experiments in (Rubanova et al., 2019), as we use a different representation of the input features. While (Rubanova et al., 2019) represents each input feature as a value-mask pair, i.e., 24 input features, we represent the data in the form of a 7-dimensional feature vector. The first four entries of the input indicate the senor ID, i.e., which arm or foot, whereas the remaining three entries contain the sensor reading.

**Event-based seqMNIST** The MNIST dataset consists of 70,000 data points split into 60,000 training and 10,000 test samples (LeCun et al., 1998). Each sample is a 28-by-28 grayscale image, quantized with 8-bits and represents one out of 10 possible digits, i.e., a number from 0 to 10.

We pre-process each sample as follows: We first apply a threshold to transform the 8-bits pixel values into binary values. The threshold is 128, on a scale where 0 represents the lowest possible and 255 the larges possible pixel value. We further transform the 28-by-28 image into a time-series of length 784. Next, we encode binary time-series in a event-based format. Essentially, the encoding step gets rid of consecutive occurrences of the same binary value, i.e., $1, 1, 1, 1$ is transformed into $(1, t = 4)$. By introducing a time dimension, we can compress the sequences from 784 to an average of 53 time-steps.

To allow an efficient batching and training, we pad each sequence to a length of 256. Note that no information was lost during this process. We normalize the added time dimension such that 256 symbols correspond to 1 second or unit of time. The resulting task is a per-sequence classification problem of irregularly sampled time-series.

**Walker2d kinematic modeling** Here we create a dataset based on the `Walker2d-v2` OpenAI gym (Brockman et al., 2016) environment and the MuJoCo physics engine (Todorov et al., 2012). Our objective is to benchmark how well the RNN architecture can model kinematic dynamical systems in an irregularly sampled fashion. The learning setup is based on an auto-regressive supervised learning, i.e., the model predicts the next state of the Walker2d environment based on the current state.

In order to obtain interesting simulation rollouts, we trained a non-recurrent policy by Proximal Policy Optimization (PPO) (Schulman et al., 2017) using the Rllib (Liang et al., 2018) reinforcement learning framework. We then collect the training data for our benchmark by performing rollouts on the `Walker2d-v2` environment using our pre-trained policy. Note that because the policy is deterministic, there is no need to include the actions produced by the policy in the training data.

We introduce three sources of uncertainty to make this task more challenging. First of all, for each rollout we uniformly sample a checkpoint of policy at 562, 822, 923, or 1104 PPO iterations. Secondly, we overwrite 1% of all actions by random actions. Thirdly, we exclude 10% of the time-steps, i.e., we simulate frame-skips/frame-drops. Note that the last step transforms the rollouts into irregularly sampled time-series and introduces a time dimension.

In total, we collected 400 rollouts, i.e., 300 used for training, 40 for validation, and 60 for testing. For an efficient training, we align the rollouts into sequences of length 64. We use the mean-square-error as training loss and evaluation metric. We train each RNN for 200 epochs and log the validation error after each training epochs. At the end, we restore the weights that achieved the best (lowest) validation error and evaluate them on the test set.

**Reproducibility statement** All code and data are submitted for review as a supplement.

Table 7: ODE-solvers used for the different RNN architectures involving ordinary differential equations

| Model | ODE-solver | Time-step ratio |
|---|---|---|
| CT-RNN | 4-th order Runge-Kutta | 1/3 |
| ODE-RNN | 4-th order Runge-Kutta | 1/3 |
| GRU-ODE | Explicit Euler | 1/4 |
| mmRNN | Explicit Euler | 1/4 |

Table 8: Hyperparameters

| Parameter | Value | Description |
|---|---|---|
| RNN latent dimension | 64 | number of neurons in the RNN |
| Minibatch size | 256 | |
| Optimizer | RMSprop (Tieleman & Hinton, 2012) | |
| Learning rate | 5e-3 | |
| Training epochs | 500/200 | Synthetic/real datasets |

