# OpenReview forum: "Mixed-Memory RNNs for Learning Long-term Dependencies in Irregularly-sampled Time Series"
_TMLR — Rejected by TMLR_

### Review · Reviewer_zG9t · 2022-04-25

**Summary Of Contributions:**

Summary of the contributions:

This paper has a theoretical part, a new learning algorithm, and experimental results. In the theory part, the authors introduce a novel definition of vanishing and exploding gradient, according to which ODE-RNNs of various kinds are guaranteed to suffer either from vanishing or exploding gradient. In the algorithm part, the mixed-memory RNN (mmRNN) is proposed, in which the state has two parts, a continuous-time state governed by a learned ODE and an LSTM-like cell state conditioned on the continuous-time state. The experiments compare the mmRNN with previous approaches on new benchmarks for data with irregular inputs.




**Requested Changes:**


I suggest to completely scrap the theory part of this paper (unless you can fix it to address the points above, but I would not know how to do it). As it is, it is not just useless (it actually does not say anything about the proposed algorithm or discriminates it with previous work), it leads to incorrect conclusions.

The experiments need to be augmented in the way suggested above in order to provide proper comparisons with existing SOTA methods for irregularly spaced temporal data.

**Strengths And Weaknesses:**


The proposed algorithm makes sense (which means some future version of this paper could be published), but there are several major weaknesses that need to be addressed for the paper to be technically correct

1. There are several problems with definition 1 (vanishing or exploding condition) at the heart of the theoretical contributions of this paper:
  1.1 The condition is going to be satisfied for almost all learning algorithms on almost all sequences and almost all time steps (depending on the choice of epsilon). This make it pretty useless in my mind. For example, the authors prove that existing ODE-RNNs satisfy the stated condition (i.e. suffer from exploding or vanishing gradient), but they say nothing about their new algorithm (mmRNN), which surely also generally satisfies the condition.
  1.2 It is quite possible for a sequence of Jacobians to not satisfy the condition and yet be without actually exploding or vanishing gradients. It suffices that the Jacobians minus the identity have opposing "signs" that make the product of Jacobians neither explode nor vanish (the successive expansions and contractions of gradient vectors cancel each other enough to avoid explosion or vanishing).
  1.3 The definition is ambiguous: is this condition to be satisfied on all data and all time steps, or any data and any time step? Clearly this will make a big difference.
  1.4 The absolute value of the sum of derivatives used in the definition is not justified anywhere. Why would it make sense to use it as a criterion in this way? I note that this sum corresponds to the directional derivative in the direction of a state-gradient vector proportional to (1,1,1,...). Why would that direction be privileged?

2. Problem 1.1 leads to "conclusions" which are gravely misleading (because the condition is likely to be satisfied for almost any reasonable RNN algorithm). According to this definition, existing algorithms like LSTMs and GRUs also suffer from vanishing-exploding gradients. It is true that they can! But this definition brings almost no information about when it happens or doesn't happen.

3. Experimental validation is mostly with homemade benchmarks. Yet, the authors cite a rich literature on algorithms for dealing with irregularly spaced inputs. Running other authors' code (or your own implementation) on these homemade benchmarks is good but by far not as convincing as running the mmRNN on EXISTING benchmarks presented in these earlier papers with results obtained by the author of these earlier papers. This needs to be done without cheating (i.e., the authors must use the state-of-the-art results on each dataset) in order to provide a convincing set of comparisons.

---

> ### Author Response · Authors · 2022-06-02
> **Response to Reviewer zG9t**
>
> We thank the reviewer for providing constructive feedback on our manuscript. Here, we comment on the issues raised:
>
> (1 and 2). We aim to restructure the paper as suggested by the reviewer to provide a continuous derivation of the gradients and discuss their effects technically, instead of the current representation of the theoretical results. We agree that almost no network/dataset pair achieves a perfect gradient condition. However, as highlighted in the paper
> "mmRNNs allow us to control the vanishing gradient compared to other continuous-time models".
>
> (3). In our revised version, we will perform and add a comparison of mmRNNs to results reported in the literature. However, we note that there exist few datasets that assess the long-term dependency learning capabilities on **irregularly sampled time series**. There are lots of sequence modeling tasks for *regularly sampled time series*, but we focus on the irregular setting. We, therefore, introduced the bit-stream classification task, as it includes long-term dependencies per definition, can be represented in a regularly sampled or irregularly sampled (run-length encoding) way, and can be made arbitrary difficult by modeling the sequence length. This
> is the first dataset that captures all three aspects.
>
> Please refer to our general response in which we describe how we will restructure the paper to address the issues raised both in the theoretical part and experimental part of the paper.

---

### Review · Reviewer_XiFC · 2022-05-13

**Summary Of Contributions:**

The paper adresses the vanishing/exploding gradient problems in the domain of some continuous-time RNNs.

To that end the computation of an ODE-RNN's gradient are explicitly represented as a the Jacobian of either Euler- or Runge-Kutta ODE solvers. The true Jacobian of the ODE is represented as the limit of a near-zero step size in a Euler setting. It is shown that the necessary and sufficient conditions for exploding and vanishing gradients can be satisfied and that recently proposed architectures do mitigate neither.

The authors further derive a condition for the absense of both exploding and vanishing gradient, which is a continuous form of the "constant error carousel" (CEC) that LSTMs feature.

Subsequently, this condition is implemented in the proposed mmRNNs, which are henceforth validated in a set of experiments where favorable performance is shown.


**Broader Impact Concerns:**

I do not see any broader impact concerns.

**Requested Changes:**

- Greatly improve the correctness of the writing: remove aforementioned errors.
- Reduction of theoretically sounding parts. Instead of summoning lemmata and theorems, make it more intuitive without sacrifice of rigour. I am certain this is possible.
- Better representation of existing theory of vanishing/exploding gradients.
- More exhaustive and transparent experimental section. Not necessarily more tasks, but clearer how HPs have been selected for the proposed and prior methods.
- Treat the adjoing method in a fairer way.


**Strengths And Weaknesses:**

The arguably greatest strength of the paper is the derivation of the architecture from first principles. Instead of proposing a heuristic, the problem of exploding/vanishing gradients is tranfered to the continuous time setting, again reduced to the discrete time setting, and then shown to be present in current architectures. I like this approach to solving problems.

Sadly, the paper comes with an array of weaknesses. I followed the proofs and discussion to related work, but as I am no expert in the recent developments of the field, my understanding may be limited.

1. I did not find the theoretical flavour of the writing appropriate, especially page 3. The insight that exploding/vanishing gradients can be present in the computation of those architectures because of the discretisation of ODE solvers, and even in the theoretical limit of near 0 step sizes, is very intuitive. I think that a series of lemmata/theorems/... is problematic and distracting.
2. The manuscript has a lot of errors, i.e. spelling mistakes. While the writing is very good for some parts, it is bad in others. This ranges from typos (" n the next section" is missing an I) over missing definite articles to wrong declinations. Also, \sigma is not defined in the main text, as far as I can tell.
3. The adjoint method is discarded in an unfair fashion. The question whether it suffers from exploding/vanishing gradients is answered hand wavily with "it can produce incorrect gradients anyway". I think it would be appropriate to say sth along the lines of "we don't know, but there are other problems which is why we don't focus on it".
4. There are two things about vanishing and exploding gradients which I found could be made more precise. For one, the condition for the exploding gradient is necessary, not sufficient. The text however makes it often sound as if it was sufficient, e.g. "we have an exploding gradient" on page 17.
   Further, architectures like LSTM cannot prevent exploding gradients, as there are additional recurrent connections in which the gradients can explode, tainting all gradients.
   Overall, this makes the impression that the theoretical foundation of the problem is not laid out well and represented in the work.
5. The experimental section has a very limited scope of hyper parameters. It is not clear if the superiority of the proposed method stems from a lucky pick of hyper parameters. How have these been tuned, and how where those of the "competing" approaches tuned? I am not convinced that the proposed method is really better.

---

> ### Author Response · Authors · 2022-06-02
> **Response to Reviewer XiFC**
>
> Thank you for providing feedback on our manuscript. Here, we comment on the issues raised:
>
> 1. We aim to restructure the paper as suggested by the reviewer with the outline presented in our general response to all reviewers. We fully agree that the current structure of the theoretical contributions of the work needs further attention, which we will do our best to address in the revised version.
>
> 2. In our revised version, we will make sure that our report is typo-free.
>
> 3. On the Adjoint method. Please denote that the gradients with stepsize 0 would vanish/explode; even if the adjoint method will be numerically correct, it would still suffer from vanishing/exploding gradient. Adjoint methods equipped with checkpointing mechanisms resolve the numerical errors induced by the Vanilla Adjoint during reverse mode computations. However, these methods would truncate the sequence and thus negatively impact the capability of models to learn long-term dependencies. We will discuss this further and will include experiments to show this in our revised manuscript.
>
> 4. We fully agree that our conditions are necessary and not sufficient. We will rework the theoretical arguments of our paper based on the feedback received from the reviewer.
>
> 5. **Hyperparameters**
> We would like to strongly emphasize that we strictly followed a very standard cross-validation hyperparameter (e.g., learning rate, etc) tuning scheme in all experiments. **Please denote that we submitted our code in full for the exact purpose of reproducibility and replicability.** Moreover, the gap in performance gain when long-term dependencies happen in irregular datasets is not incremental:
>
> - Please see Table 3 3rd column, where the only model that is marginally close to our method is GRU-D [Che et al. 2018]. A large body of continuous processes such as ODE-RNNs, CT-RNNs, CT-GRU, RNN-decay, GRU-ODE, coRNN, and Lipschitz RNN including iRNNs could not even learn meaningful representations on this task.
>
> - In Table 5, mmRNN gained 2.9% improved performance compared to the 2nd, 3rd, and, 4th best models. That means 12 standard deviations (std) better than Lipschitz RNNs, 7 std better than GRU-D, and 14 std better than coRNN. The gap in performance vs these models is with high confidence significantly significant.
>
> - Finally, in Table 6, where the task includes missing data (deleted sample points), increased uncertainty (replacements with some random actions), long-term dependencies, and irregularly sampled, we see that mmRNNs significantly outperform all models with a large margin.
>
> We will certainly include a section in our revised manuscript about hyperparameter tuning and how it is performed in our case.

---

### Review · Reviewer_fkfY · 2022-05-14

**Summary Of Contributions:**

This paper first provides formal study on the gradient vanishing/explosion problem in the RNN framework. The authors provide the analysis showing that the existing ODE-RNN frameworks would suffer from these problems when modeling irregularly-sampled time series, regardless of what ODE-solver being used. Then the authors proposed a new mmRNN which is claimed to be able to control the vanishing gradients. The core idea is to have the hidden states change between time stamps with continuous-time ODE, while having a discrete-time controller to instrument the ODE solving. Experiments on some synthetic and real-world datasets show that the proposed approach could achieve better performance, compared to other RNN or ODE-RNN alternatives.


**Requested Changes:**

Please see the 'cons' section above for the requested changes. The corresponding significances are as follows:

- The empirical justification is not significant enough: major

- The technical details are not well explained: important

- Correctness of the proposed approach: important

- Some presentation issues: minor


**Strengths And Weaknesses:**

Pros:

- The analysis of ODE-RNN on the topic of gradient vanishing/explosion problem is interesting and important.
- The proposed approach is simple yet effective.
- Experiments show some empirical gains.

Cons:

- The empirical justification is not significant enough.
1) Firstly, as the claim of the paper is to solve the gradient vanishing/explosion problem, it would be necessary to justify this directly, rather than using the classification/regression precision as indirect measurements. For example, having a setting that causes the vanishing/explosion problem of existing ODE-RNN while showing the mmRNN is robust would be necessary.

2) Secondly, the sequence classification task might not be informative / appropriate enough to demonstrate the advantage of the proposed approach, as it is not clear to me why the mmRNN is supposed to be more expressive, or generalize better than existing methods.

3) There are no strong Transformer based approaches involved in the comparison. As it has been shown in many time series forecast problems (e.g., Informer: Beyond Efficient Transformer for Long Sequence Time-Series Forecasting, AAAI 2021) that Transformers are showing better results, it would be good to see the comparisons here as well.

- The technical details are not well explained.

1) Firstly the ODE-RNN is only briefly introduced in Sec 2. It would be necessary to incorporate the details, including parameterization, optimization approaches, etc.

2) It seems the f_{\theta} in Eq (4) is never explained.

- Correctness of the proposed approach.

It would be great if the authors can provide more details of why the mmRNN would get rid of the issue completely. Specifically, as the authors mentioned in Theorem 1 where ODE-RNN would suffer from the explosion/vanishing problem, it is not clear how it would be affected by the number of discretized points in the time span. Suppose the time span is large, then would it still suffer to integrate over that large time span?

- Some presentation issues:

In Eq (9) in appendix, what is the n and m? As it is mentioned above that n = 1, then it makes Eq (9) really confusing.

---

> ### Author Response · Authors · 2022-06-02
> **Response to Reviewer fkfY**
>
> Thanks for providing feedback on our manuscript. Here, we comment on the issues raised:
>
> 1. Table 2 of the manuscript (pasted here) shows empirical evidence that mmRNN strongly controls the vanishing/exploding gradient problem, compared to ODE-RNNs. *Empirical measurement of gradient norms* To emphasize the impact of our theoretical results, we performed an experiment comparing the hidden state gradient norms of an ODE-RNN instance compared to another ODE-RNN equipped with mmRNN for different sequence lengths (mean/std over 3 initialization seeds). The results show (as proven in Theorem 1 and 2) that the gradients of a standard ODE-RNN tend to exponentially increase with the length of the input sequence. This gradient issue is significantly improved when mmRNNs are used.
>
> |  Sequence length  |  5  |  10  |  25  |  50  |
> |---|---|---|---|---|
> | ODE-RNN |10.50 $\pm$ 5.47 | 38.75 $\pm$ 33.20 | 235.76 $\pm$ 383.04 | 1214.94 $\pm$ 3750.62 |
> | mmRNN |0.97 $\pm$ 0.01 | 0.96 $\pm$ 0.03 | 1.01 $\pm$ 0.10 | 1.25 $\pm$ 0.25 |
>
> 2. We argue that there exist few datasets that assess the long-term dependency learning capabilities on **irregularly sampled time series**. There are lots of sequence modeling tasks for regularly sampled time series, but we focus on the **irregular setting**. We, therefore, introduced the bit-stream classification task, as it exactly encodes this setting, i.e., includes long-term dependencies per definition, can be represented in a regularly sampled or irregularly sampled (run-length encoding) way, and can be made arbitrary difficult by modeling the sequence length. This is the first dataset that captures all three aspects. **Our experiments indicate that for **all** models, learning long-term dependencies in irregularly sampled is more difficult than for regular sequences.**
>
> 3. Transformer models perform greatly on sequence modeling tasks *as long as they have been pre-trained on a large corpus of data first*. If we train transformer models from scratch on the exact dataset we used in all our experiments, they would not perform as well as the reported baselines. As we plan to restructure our paper (explained in our general comments to all reviewers) we will include some Transformer benchmarks as well.
>
> Nevertheless, please denote that Transformers compute pair-wise (query-key) values for all possible pairs in a sequence and therefore circumvent the vanishing gradient problem at the cost of quadratic computational complexity. However, this work is about improving existing continuous-time RNN architectures by equipping them with a memory component. We will add the technical details about this matter in our revised manuscript.
>
> **Technical descriptions and Correctness**
>
> mmRNN does not resolve the vanishing/exploding gradient completely as stated theoretically and experimentally in our paper (see above table). mmRNN significantly improves this gradient issue of continuous-time models.
>
> Moreover, as briefly mentioned in Theorem 1, a perfect constant error propagation might be detrimental to learning useful dynamics. For example, an ODE with zero dynamics has a perfect error transport according to the common
> definitions of vanishing/exploding gradients, but cannot implement useful functions. Therefore, empirical results on real datasets are necessary to assess the performance of an RNN model.

---

### Author Response · Authors · 2022-06-02
**General Response to Reviewers and Editors**

We would like to thank all reviewers for their constructive feedback on our manuscript. We are very excited to have received feedback that can truly shape this work towards a great publication, and for that, we thank the TMLR Team.

Based on the reviewers’ suggestions, we aim to restructure our work with the following agenda:

1. We will rework the theoretical section of our paper to precisely reflect the necessary and sufficient conditions under which our claims hold. In particular, we will first provide an intuitive description of the problem at hand and its definition. We will then articulate with mathematical rigor how ODE-based methods trained via reverse mode automatic differentiation, suffer from vanishing/exploding gradients, and how mmRNNs can **control** the effect and not resolve the effect.

2. We will extend the experimental results of our paper to additional standard benchmarks on irregularly sampled data with long-term dependencies. In particular, as suggested by the reviewer, we will additionally include new baseline models on the topic to emphasize the impact of our mixed-memory architectures.

**For incorporating these revisions we would like to ask for 10-12 weeks of time.** We would greatly appreciate if the Editor and the reviewers agree to this time frame.

---

### Decision · Action_Editors · 2022-06-19

**Recommendation:** Reject

**Comment:**

*Summary:*
This paper looks into the critical problem of dealing with irregularly sampled time-series problems that show up in several real-world problems. Typical deep learning-based sequence modeling algorithms can't deal well with the irregularly sampled time series. Recently proposed ODE-RNNs try to address this. However, this paper shows that ODE-RNNs can suffer exploding and vanishing gradient problems. They propose the mmRNN approach to address the vanishing gradient problem with ODE-RNNs on continuous time-series tasks. The authors provide some theoretical justification for their approach and then try to support their theory with the empirical results.


*Reviewer's feedback:*
Overall the reviewers acknowledged that the problem this paper is trying to address is essential, and they found the proposed solution interesting.

- However, reviewers fkfY, XiFC, and zG9t all argued that the experimental results in the paper are insufficient. Reviewer zG9t complained that the experiments are mostly on homemade benchmarks. Reviewer XiFC mentioned that the experiments are not exhaustive and transparent enough; they have a minimal scope of hyper-parameters. Reviewer fkfY didn't find the experimental results significant enough and suggested changes to improve them.

- Reviewers fkfY and XiFC didn't find the writing of the paper clear enough, and they both suggested several improvements.

- Reviewer zG9t and XiFC pointed out severe concerns and flaws in the theory. Reviewer zG9t recommended completely scrap the theory part of this paper

*Recommendation:*

The reviewers have provided detailed revisions and unanimously recommended this paper for rejection. I agree with the reviewers' justifications and recommendations. The authors acknowledged the suggested edits and requested 10-12 weeks to revise the paper. Since this is a very long time for revision, I would recommend the authors revise the paper and resubmit it to TMLR when it is ready.